# DepthVanish: Optimizing Adversarial Interval Structures for Stereo-Depth-Invisible Patches

**Yun Xing**[1,2,5*]   **Yue Cao**[5,6*]   **Nhat Chung**[5]   **Jie Zhang**[5]   **Ivor Tsang**[5,6]
**Ming-Ming Cheng**[2,3]   **Yang Liu**[6]   **Lei Ma**[1,4]   **Qing Guo**[2,5†]
[1] University of Alberta, Canada    [2] VCIP, CS, Nankai University, China
[3] NKIARI, Shenzhen Futian, China    [4] The University of Tokyo, Japan
[5] CFAR and IHPC, Agency for Science, Technology and Research (A*STAR), Singapore
[6] Nanyang Technological University, Singapore

## Abstract

Stereo depth estimation is a critical task in autonomous driving and robotics, where inaccuracies (such as misidentifying nearby objects as distant) can lead to dangerous situations. Adversarial attacks against stereo depth estimation can help reveal vulnerabilities before deployment. Previous works have shown that repeating optimized textures can effectively mislead stereo depth estimation in digital settings. However, our research reveals that these naively repeated textures perform poorly in physical implementations, *i.e.*, when deployed as patches, limiting their practical utility for stress-testing stereo depth estimation systems. In this work, for the first time, we discover that introducing regular intervals among the repeated textures, creating a grid structure, significantly enhances the patch's attack performance. Through extensive experimentation, we analyze how variations of this novel structure influence the adversarial effectiveness. Based on these insights, we develop a novel stereo depth attack that jointly optimizes both the interval structure and texture elements. Our generated adversarial patches can be inserted into any scenes and successfully attack advanced stereo depth estimation methods of different paradigms, *i.e.*, RAFT-Stereo and STTR. Most critically, our patch can also attack commercial RGB-D cameras (Intel RealSense) in real-world conditions, demonstrating their practical relevance for security assessment of stereo systems. The code is officially released at: `https://github.com/WiWiN42/DepthVanish`

## 1 Introduction

Depth estimation is a crucial component in safety-critical embodied systems like autonomous driving [6] and robotics [3], where accurate perception of the 3D environment is essential for reliable operation. Investigating the errors in depth estimation, such as mistaking nearby objects as distant ones in safety-critical embodied systems [30, 5, 38, 8, 20, 42, 41], can provide critical insights for safety practices. Most existing works focus on the security vulnerabilities of monocular depth estimation, which relies heavily on scene priors from single images. Stereo depth estimation, on the other hand, utilizes geometric constraints and typically provides more robust and metrically accurate results, making it attractive for high-stakes applications.

However, despite this inherent advantage, recent studies revealed that DNN-based stereo pipelines remain vulnerable to adversarial attacks, as carefully crafted pixel-level perturbations [31, 1] can cause substantial disparity estimation errors. Nevertheless, previous works have primarily addressed digital

---

*indicates equal contribution. This work was done during Yun Xing was an intern at CFAR & IHPC, A*STAR and Nankai University. † Corresponding author, email: tsingqguo@ieee.org.

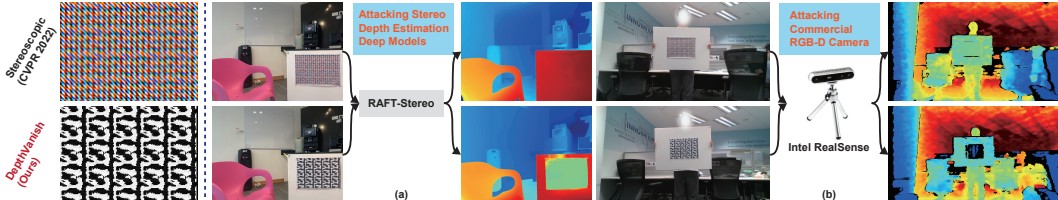

**Figure 1:** Baseline (Stereoscopic [1]) *vs.* our DepthVanish on attacking RAFT-Stereo [19] and Intel RealSense.

attacks utilizing full-image noise, which are impractical in real-world contexts due to constraints like limited patch size, varying viewing angles, and dynamic lighting conditions, *etc.* As illustrated in the first row of Fig. 1, when applied as a physical patch, the existing Stereoscopic [1] fails to effectively attack RAFT-Stereo and Intel RealSense. This lack of physically realizable and generalizable attack methods presents a significant limitation in evaluating the robustness of stereo systems, particularly as stereo estimation continues to be deployed in real-world, safety-critical applications.

In this study, we address these limitations by introducing the first adversarial patch attack that is effective in both digital and physical settings against widely deployed deep stereo depth estimation models (Fig. 1 second row). Fundamentally, we discover that adding regular intervals among repeated textures to form a spatial structure shows great potential for improving the attack effectiveness and enables digital-to-physical transferability. Through systematic analysis, we show how interval spacing influences the attack success. These insights inform a novel optimization pipeline that jointly designs patches' texture and structure to achieve high attack effectiveness across models and deployment settings. Thus, we propose a novel optimization pipeline that co-designs both texture elements and interval structure for generating adversarial patches that ❶ remain effective when physically printed and inserted into real scenes, ❷ work across diverse datasets and environments and ❸ generalize across different stereo depth estimation models, including commercial RGB-D sensors, *i.e.*, Intel RealSense. In summary, our contributions are as follows,

- We introduce the first adversarial attack that is both digitally and physically effective for deep stereo estimation models including the advanced RAFT-Stereo and Stereo Transformer.
- By conducting a comprehensive empirical study, we discover that regular interval spacing among repeated textures significantly improves the patch attack effectiveness and its real-world transferability over naive texture repetitions.
- We develop a joint optimization algorithm, *i.e.* DepthVanish, that co-designs the texture and its spatial structure within the patch to maximize the digital and physical attack effectiveness.
- By physically evaluating our patch, we expose severe safety concerns of existing stereo depth estimation systems and highlight the emergency of practical model robustness enhancement.

## 2 Related Work

**Stereo depth estimation.** Stereo-based depth estimation is a technique that infers scene depth from visual correspondences, which captured as disparity maps, between pairs of stereo images in various applicable settings [23, 35, 2, 25, 24, 15]. Traditional methods typically follow a multi-stage pipeline involving the computation of matching costs, cost aggregation, and optimization to predict and refine disparities [26, 4, 33, 10]. In contrast, recent advances have incorporated deep neural networks [29], enabling end-to-end learning of feature representations for correspondence matching and direct prediction of disparity and/or depth. In particular, CNN-based methods [4, 12, 34, 39, 22, 27] typically build 3D cost volumes from shared-weight feature encoders, attention-based models [18, 11, 28, 14] employ vision transformers to model global correspondences and disambiguate difficult regions, and iterative refinement methods [19, 16, 32] apply a recurrent update operator to progressively converge on the final disparity, avoiding the memory-intensive 3D cost volumes. Compared to monocular methods [36, 37], stereo offers improved robustness by leveraging geometric constraints from dual viewpoints, but still faces challenges in low-texture and repetitive areas [40, 21].

**Depth estimation attack.** Due to their effectiveness and capability for real-time performance, depth estimation systems have become essential components of safety-critical applications such as autonomous driving [6] and robotic navigation [3]. Monocular depth estimation models, in particular, have been extensively studied under both digital [30, 5, 38] and physical adversarial

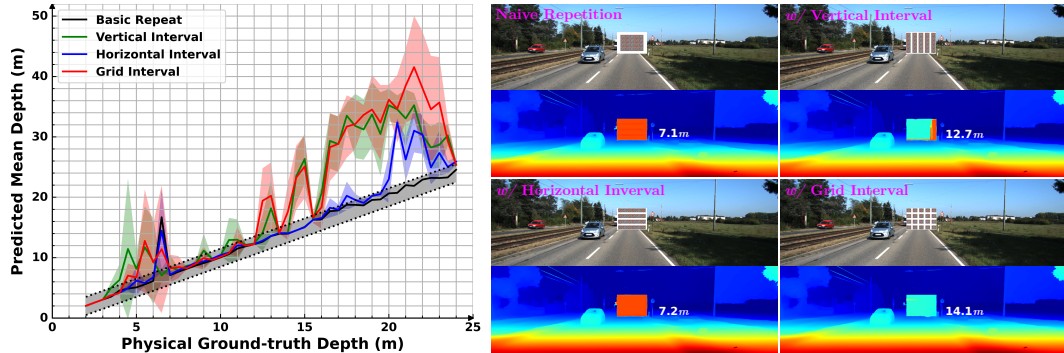

(a) Performance of Different Spacing Strategy     (b) Visualization of Different Spacing Strategy

**Figure 2:** Adversarial effect of interval spacing on depth prediction. (a) Mean predicted depth (solid lines) and variance (shaded regions) for different interval spacing strategies, averaged over interval widths of $2 - 10\ px$. The gray dashed band indicates $\pm 1.5\ m$ from the ground truth. (b) Visualization of depth prediction results for typical different interval spaced patches where the ground truth depth is $7\ m$.

attacks [8, 20, 42, 41]. These evaluations have revealed various system vulnerabilities and led to the development of tailored defense strategies [13, 9, 7], including adversarial training and robust feature learning. In contrast, despite their geometric soundness and widespread deployment, stereo depth estimation systems [19, 17] have received limited attention in adversarial research. Existing research has focused primarily on digital, white-box attacks [1, 31], overlooking potential vulnerabilities in the physical world. This gap is particularly concerning, as stereo systems rely on precise correspondence between left and right images. Failures in such systems can lead to serious consequences, especially in autonomous applications where accurate and reliable 3D perception [35] is critical.

## 3 Motivation

### 3.1 Naive Repetition Fails in Realistic Patch Attacks

Stereo depth estimation recovers 3D structure by identifying correspondences between left and right images [43], typically formulated as a pixel-wise optimization along epipolar lines:

$$d^*(x) = \arg\min_d C(x, d), \tag{1}$$

where $d \in \mathbb{Z}$ represents the horizontal disparity between pixel $x$ in the left image and pixel $x - d$ in the right image, and $C(x, d)$ denotes the matching cost between them. When repetitive patterns are presented, the cost volume exhibits periodic ambiguity [26]:

$$C(x, d) \approx C(x, d + ns), \quad \forall n \in \mathbb{Z}, \tag{2}$$

where $s$ denotes the spatial repetition period. This periodicity produces multiple equally plausible matches, thereby increasing the likelihood of incorrect or unstable depth estimations.

Previous adversarial attacks [1, 31] inject repetitive optimized noise over the entire image to exploit such periodic ambiguities. Since global injection is impractical in real-world scenarios, we instead explore attacks using localized adversarial patches. As shown by the black curve in Fig. 2(a), we deploy the repetitive noise from [1] as patch into a real-world scene at different ground-truth depth and plot the corresponding predicted mean depth. It can be seen that simply repeating the noise within patches results in predicted depth that remains same to the ground truth, indicating limited adversarial effectiveness. This is visually confirmed in Fig. 2(b) (top left), where a naive repeated patch yields a predicted depth of $7.1\ m$, which is almost identical to the ground truth of $7\ m$. This observation reveals a key limitation of existing studies: naive repetition fails to generate sufficient ambiguity within practical patches, which motivates the need for more structured pattern designs.

### 3.2 Structured Intervals: Enhancing Patch Adversarial Effectiveness

To address the limitation, we propose introducing regular intervals into the repetitive pattern to amplify the matching ambiguity as Eq.(2), thereby enhancing the adversarial effect of the patch.

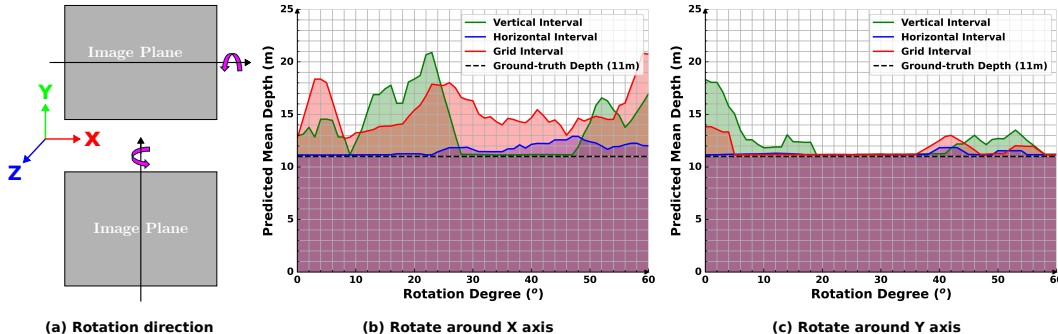

**Figure 3:** RAFT-Stereo depth prediction performance under various interval structures and patch rotation degrees. (a) Illustration of rotation around the X and Y axes. (b) Depth prediction performance at different rotation degrees around X axis. (c) Depth prediction performance at different rotation degrees around Y axis.

As demonstrated in Fig. 2(b), given the patch with basic repetitive pattern from [1] (top left), we add vertical (top right), horizontal (bottom left) and grid (bottom right) space to form patches with structured intervals. We systematically evaluate the impact of different interval configurations on the RAFT-Stereo model using the KITTI dataset. As shown in Fig. 2(a), structured intervals notably enhance attack effectiveness. ❶ Basic Repeat (**black**): the predicted patch depth remain close to the ground truth depth indicating minimal adversarial influence, which is also verified by the visualization in Fig. 2(b) (top left). ❷ Horizontal Interval (**blue**): moderate overestimation beyond $15\ m$ (*e.g.*, a $20\ m$ true depth yields a $\sim 24\ m$ prediction). Visual results (Fig. 2(b), bottom left) confirm a slight increase to $7.2\ m$. ❸ Vertical Interval (**green**): produces larger errors, frequently reaching $\sim 30\ m$ at a $20\ m$ ground truth. In Fig. 2(b) (top right), the predicted depth surges to $12.7\ m$. ❹ Grid Interval (**red**): combining intervals in both directions produces the strongest adversarial effect, with depth predictions surpassing $40\ m$ at a $23\ m$ ground truth. In the visual result (Fig. 2(b), bottom right), the predicted depth reaches $14.1\ m$, demonstrating a significant adversarial effectiveness.

In summary, structuring the patch with both horizontal and vertical intervals (*i.e.*, grid spacing) greatly increases the adversarial effect of patches, far exceeding the impact of simple repetition. However, we also observe two critical limitations: ❶ the overall attack performance remains limited, especially when the patch is placed close to the camera. ❷ the significant variation in performance across different interval configurations suggests that a single fixed interval structure is insufficient.

### 3.3 Structured Intervals: Improving Attack Robustness across Viewpoints

A practical adversarial patch must maintain its effectiveness even when the patch is rotated or viewed from different orientations. This is particularly important under real-world deployment conditions, where precise placement is difficult to control. To this end, we systematically evaluate the impact of interval structure on attack robustness against patch rotation. As shown in Fig. 3(a), we rotate the patch along two axes (*i.e.*, X and Y) and summarize the predicted mean depth in Fig. 3(b) and (c).

For X-axis rotation (Fig. 3(b)): ❶ the horizontal (**blue**) and vertical (**green**) intervals exhibit angle-dependent performance, succeeding only at certain angles; ❷ the grid interval (**red**) is more robust, demonstrating more consistent effectiveness across different angles. For Y-axis rotation (Fig. 3(c)), although all configurations show moderate attack robustness across viewpoints, adding intervals still yields improvements. These results show that structured intervals improve attack robustness to patch rotation, which is essential for reliable adversarial attacks in real-world scenarios.

In summary, the above findings underscore the promise of structured intervals but also reveal their limitations under varying depths and configurations. These observations highlight the need for further optimization of the patch's texture and structure to achieve more effective attacks.

## 4 Problem Formulation

To formalize the stereo depth estimation task and define our attack objective, we begin with the following setup. Given a stereo image pair $(\mathbf{I}_l, \mathbf{I}_r)$ where $\mathbf{I}_l, \mathbf{I}_r \in \mathbb{R}^{3 \times H \times W}$ of a specific scene, a pretrained stereo depth estimation model $\mathcal{F}(\cdot)$ predicts the pixel-wise disparity map $\mathbf{d}_{pred} =$

$\mathcal{F}(\mathbf{I}_l, \mathbf{I}_r) \in \mathbb{R}^{H \times W}$. The corresponding depth map is computed as $\mathbf{z} = \frac{f \times B}{\mathbf{d}_{pred}}$ where $f$ and $B$ denote the focal length and baseline of the stereo camera rig respectively.

In general, the objective of adversarial patch attack is to construct a patch $\mathbf{P} \in \mathbb{R}^{3 \times h_p \times w_p}$ such that the stereo depth estimation model $\mathcal{F}(\cdot)$ produces an incorrect disparity output for the patch:

$$\mathcal{F}^p(\hat{\mathbf{I}}_l, \hat{\mathbf{I}}_r) \neq \mathcal{F}^p(\mathbf{I}_l, \mathbf{I}_r) \tag{3}$$

where $\hat{I}_l$ and $\hat{I}_r$ denote the stereo images with the adversarial patch $\mathbf{P}$, and $p$ indicates the corresponding pixel region occupied by the patch within the prediction results. As analyzed in Sec. 3, interval spacing can trigger critical depth estimation failures, *i.e.*, the disappearance attack. To expose the severity of such vulnerability, we define a more destructive attack objective:

$$\mathcal{F}^p(\hat{\mathbf{I}}_l, \hat{\mathbf{I}}_r) = \mathbf{0}, \quad s.t. \ \mathbf{d}_{gt}^p = \mathbf{c}, \tag{4}$$

where the model predicts zero disparity for the patch region (*i.e.*, infinite depth), despite the ground truth disparity of the patch, $\mathbf{d}_{gt}^p$, indicating a fixed, close distance $(f \times B)/\mathbf{c}$. This attack objective reveals more severe vulnerabilities than Eq. (3) and poses substantial safety risks, particularly when the patch is physically realizable and effective in real-world deployments.

# 5 Methodology

In this work, we build upon our novel findings in Sec. 3 and propose realizing the attack goal in Eq. (4) by exploiting the attack capability of interval spacing. However, this is a non-trivial problem since ❶ Eq. (4) requires the patch's ground-truth depth to be close but Fig. 2 (a) indicates that interval spacing exerts only a limited adversarial effect when the patch is deployed closely. Moreover, ❷ the robustness against rotation is a critical requirement for the patch to be physically attack effective. Yet we observed in Fig. 3 that the robustness provided by the naive interval strategy is rather limited especially against the rotation of Y axis. As a result, it is obvious that an advanced interval spacing strategy is required to realize our attack goal as defined in Eq. (4).

Fundamentally, interval spacing induces a mask $\mathbf{M}$ that partitions the patch $\mathbf{P}$ into interval structure $\mathbf{P}_s = \mathbf{M} \odot \mathbf{P}$ and texture content $\mathbf{P}_t = (1 - \mathbf{M}) \odot \mathbf{P}$, such that $\mathbf{P} = \mathbf{P}_s + \mathbf{P}_t$. Hence, we propose to optimize these components to reveal their adversarial effects. Beginning with the naive interval spacing strategy, and thus the mask $\mathbf{M}$, in Sec. 3, we first focus on optimizing the texture content $\mathbf{P}_t$, which composed of tiled texture elements $\mathbf{E}$, forming the basis of our Grid-based Attack. We then introduce the DepthVanish Attack, which jointly optimizes both $\mathbf{P}_s$ and $\mathbf{P}_t$ for maximal effect.

## 5.1 Grid-based Attack

In general, it is straightforward to setup an optimization pipeline for optimizing the texture element with grid intervals, where the patch is formed by repeating the texture elements over the grid. Fundamentally, there two main aspects that need to be considered: ❶ the physical constraint required for the texture element to form a patch and ❷ the objective function adopted for optimization.

Given our primary goal is to achieve physical attack effectiveness, the patch must comply with the physical geometry constraints during the optimization. Specifically, given a user pre-defined physical patch size $(u, v)$ and physical distance to the camera $e$ in meters, we first find the corresponding pixel size of the patch $(h_p, w_p)$ with the help of stereo calibration information (See details in Sec. 6.1). Then, we empirically adopt the optimal interval width $o$ and number of repetition $k$ from Sec. 3 to determine the texture element $\mathbf{E}$ size $(h_t, w_t)$ as

$$h_t = \frac{h_p - k \cdot o}{k + 1}, \quad w_t = \frac{w_p - k \cdot o}{k + 1}. \tag{5}$$

Based on the size of the texture element, the texture component $\mathbf{P}_t$, and consequently the full patch $\mathbf{P}$, is constructed by tiling the base texture unit $\mathbf{E}$ in a regular grid pattern as illustrated in Fig. 2(b) (bottom right). With the correctly assembled and deployed grid-based patch, we set the optimizing objective function as regional mean square error (rMSE) which is formulated as

$$\mathcal{L}_{rMSE} = \frac{1}{h_p \cdot w_p} \sum_{i=1}^{h_p} \sum_{j=1}^{w_p} (\mathcal{F}(\hat{\mathbf{I}}_l, \hat{\mathbf{I}}_r) - \mathcal{F}(\mathbf{I}_l, \mathbf{I}_r))^2. \tag{6}$$

Let $\mathcal{R}$ be the set of $k \times k$ grid locations on where $\mathbf{E}$ is repeated. The texture element is updated with average gradients: $\mathbf{E} \leftarrow \mathbf{E} - \eta \cdot \frac{1}{|\mathcal{R}|} \sum_{(i,j) \in \mathcal{R}} \nabla_{\mathbf{E}} \mathcal{L}_{rMSE}^{(i,j)}$, where $\eta$ is the learning rate. Gradients are only applied to the repeated texture regions while the interval areas remain untouched.

## 5.2 DepthVanish Attack

As we will see in Fig. 5, the above grid-based optimization can successfully mount an attack against various stereo systems but the results are still far from our attack goal defined in Eq. (4). Thus we further consider optimizing the interval structure $\mathbf{P}_s$ simultaneously during the updating of the texture element $\mathbf{E}$. Practically, optimizing the interval structure on patch level will break the texture repetitions as the interval will be updated to have irregular size. To keep the repetitions and incorporate the interval's attack capability, we propose to jointly optimize the interval structure within the texture element and, following [1], tile the optimized texture element $\mathbf{E}$ to form the final patch.

Same to grid-based attack, given a user pre-defined patch physical size $(u, v)$ and physical distance to the camera $e$ in meters, we first find the corresponding pixel size of the patch $(h_p, w_p)$. Then we calculate the texture element size $(h_t, w_t)$ by simply dividing $(h_p, w_p)$ to the repetition times $k$. In order to optimize the texture element so that the interval structure integrated as part of the texture, we propose to regularize the texture element $\mathbf{E}$ during optimization with two objectives. First, we directly cast entropy constraint on the texture element for regularizing its values to be binary, so that a crisp separation is formed to serve as the required interval structure:

$$\mathcal{L}_{entropy} = \frac{1}{h_t \cdot w_t} \sum_{i=1}^{h_t} \sum_{j=1}^{w_t} -\mathbf{E}_{ij} log(\mathbf{E}_{ij} + \epsilon) - (1 - \mathbf{E}_{ij}) log(1 - \mathbf{E}_{ij} + \epsilon). \tag{7}$$

However, we experimentally found that the texture element cannot form a clear pattern with only entropy regularization. As a result, we further integrate the total variation loss to penalizes local pixel-level variation, encouraging the formation of smooth areas:

$$\mathcal{L}_{tv} = \frac{1}{h_t \cdot w_t} \sum_{i=1}^{h_t} \sum_{j=1}^{w_t} |\mathbf{E}_{i+1,j} - \mathbf{E}_{ij}| + |\mathbf{E}_{i,j+1} - \mathbf{E}_{ij}|. \tag{8}$$

With the entropy and total variant constraints, we arrived at an objective function that can shape a clearly interval pattern for the texture element. In summary, the overall objective function adopted for optimization is formulated as

$$\mathcal{L} = \mathcal{L}_{rMSE} + \alpha * \mathcal{L}_{entropy} + \beta * \mathcal{L}_{tv}, \tag{9}$$

where $\alpha$ and $\beta$ are the hyper-parameters balancing the sharp border and coherent region requirements. Hence, we update with $\mathbf{E} \leftarrow \mathbf{E} - \eta \cdot \frac{1}{|\mathcal{R}|} \sum_{(i,j) \in \mathcal{R}} \nabla_{\mathbf{E}} \mathcal{L}^{(i,j)}$ where $\eta$ is the learning rate.

### 5.3 Implementation Details.

During the optimization for the Grid-based and DepthVanish attacks, we use a patch with a physical size $(u, v) = (0.891\,m, 1.26\,m)$ and specify the physical ground-truth depth $e = 5\,m$. To assemble the texture element into a patch, we empirically set the number of repetition as 5 for horizontal and 4 for vertical, i.e., $k = (4, 5)$. For the optimization and corresponding evaluation results with different patch physical setup, we provide them in the supplemental material. When the patch is optimized as grid-based attack, the optimal interval size $o = 10\,px$ from Sec. 3 is applied. As for the loss weights adopted during the depth vanish attack, we keep setting $\alpha = 0.1$ and $\beta = 10$. Please find more details of implementation for both Grid-based and DepthVanish attack in the supplemental material.

## 6 Experiments

### 6.1 Experimental Setup

**Dataset.** For the evaluation of digital attack effectiveness, we adopt the stereo images from KITTI scene flow (KITTI-scene) [23] and DrivingStereo [35] datasets. Both datasets are composed of stereo images of urban traffic scenes where the image size of KITTI-scene is (1242, 375) and DrivingStereo

**Table 1:** Statistical attack performance of our DepthVanish, grid-based patch and existing baselines for PSMNet, DeepPruner, AANet, RAFT-Stereo and STTR on KITTI-scene dataset. The best results are highlighted in **bold**.

| KITTI-scene | PSMNet | | DeepPruner | | AANet | | RAFT-Stereo | | STTR | |
|---|---|---|---|---|---|---|---|---|---|---|
| | D1 | EPE | D1 | EPE | D1 | EPE | D1 | EPE | D1 | EPE |
| Stereoscopic Patch | $6.23_{\pm1.13}$ | $5.28_{\pm0.88}$ | $8.29_{\pm10.23}$ | $3.29_{\pm2.05}$ | $6.79_{\pm2.30}$ | $3.69_{\pm0.39}$ | $5.79_{\pm9.88}$ | $3.58_{\pm6.73}$ | $4.58_{\pm2.83}$ | $1.30_{\pm5.37}$ |
| Stereopognosia Patch | $2.17_{\pm0.09}$ | $2.18_{\pm0.58}$ | $5.40_{\pm11.16}$ | $1.62_{\pm2.33}$ | $3.42_{\pm2.59}$ | $1.96_{\pm0.44}$ | $4.18_{\pm11.27}$ | $2.09_{\pm12.66}$ | $3.02_{\pm3.00}$ | $1.28_{\pm8.79}$ |
| Grid-based Patch (ours) | $3.35_{\pm1.09}$ | $48.21_{\pm8.24}$ | $55.39_{\pm10.77}$ | $38.60_{\pm3.03}$ | $60.59_{\pm8.90}$ | $53.84_{\pm4.93}$ | $40.09_{\pm5.87}$ | $\mathbf{67.24_{\pm7.90}}$ | $5.23_{\pm7.49}$ | $45.34_{\pm7.30}$ |
| **DepthVanish (ours)** | $\mathbf{55.30_{\pm6.85}}$ | $\mathbf{50.71_{\pm9.71}}$ | $\mathbf{97.07_{\pm12.42}}$ | $\mathbf{67.19_{\pm4.85}}$ | $\mathbf{66.42_{\pm10.10}}$ | $\mathbf{56.54_{\pm5.26}}$ | $\mathbf{89.31_{\pm6.56}}$ | $66.01_{\pm6.18}$ | $\mathbf{92.38_{\pm8.76}}$ | $\mathbf{69.25_{\pm6.62}}$ |

**Figure 4:** Attack performance of our DepthVanish, grid-based patch and existing baselines for PSMNet (M1), DeepPruner (M2), AANet (M3), RAFT-Stereo (M4) and STTR (M5) on the sub-sets of DrivingStereo dataset.

is (1758, 800). In more detail, we adopt the four sub-sets of DrivingStereo that were captured under different weather conditions (*i.e.*, sunny, foggy, rainy, cloudy) where we report the attack performance for each of them respectively. Following [1], 40 stereo image pairs for each (sub-)dataset are selected to verify the effectiveness of different patches. For the physical evaluation, we manually capture stereo images with i3DStreoid [2] where various safety critical situations are considered. We refer readers to the supplemental material for the details of how the physical stereo images are captured and the pipeline we adopted for physical deployment.

**Attack targets.** Following [31, 1], we apply our attack method to PSMNet [4], DeepPruner [10] and AANet [33] for validating the general attack effectiveness. Moreover, we empirically found that they are out-of-date and can be easily disturbed, thus we further select RAFT-Stereo [19] and STereo TRansformer (STTR) [17] which represent the promising iterative optimization-based methods and transformer-base methods as our main attack targets. For the detail hyper-parameter setting and the pretrained checkpoint adopted during the attack, please find all of them in supplemental material.

**Digital deployment.** During the digital optimization and evaluation, the patch needs to be placed inside the scene according to physical constraints. To achieve this, we apply the calibration information provided by the KITTI and DrivingStereo dataset. In specific, given a patch with a predefined physical size in meters, we first set the homogeneous 3D coordinates of the patch's corners with respect to the reference camera coordinate system. Then we calculate the corresponding pixel coordinates with the help of the rectified projection and rotation matrix. The full calculation is detailed in supplement.

**Evaluation metrics.** Following the convention, we adopt bad pixel error (D1-error) and End-Point Error (EPE) for evaluating the prediction performance which are calculted as follows:

$$\text{D1} = \frac{\text{\# of bad pixels}}{\text{\# of total pixels}} \times 100\%, \qquad \text{EPE} = \frac{1}{N}\sum_{i=1}^{N}|\mathbf{d}_{pred}^{i} - \mathbf{d}_{gt}^{i}|, \tag{10}$$

---

[2]http://stereo.jpn.org/eng/iphone/help/index.html

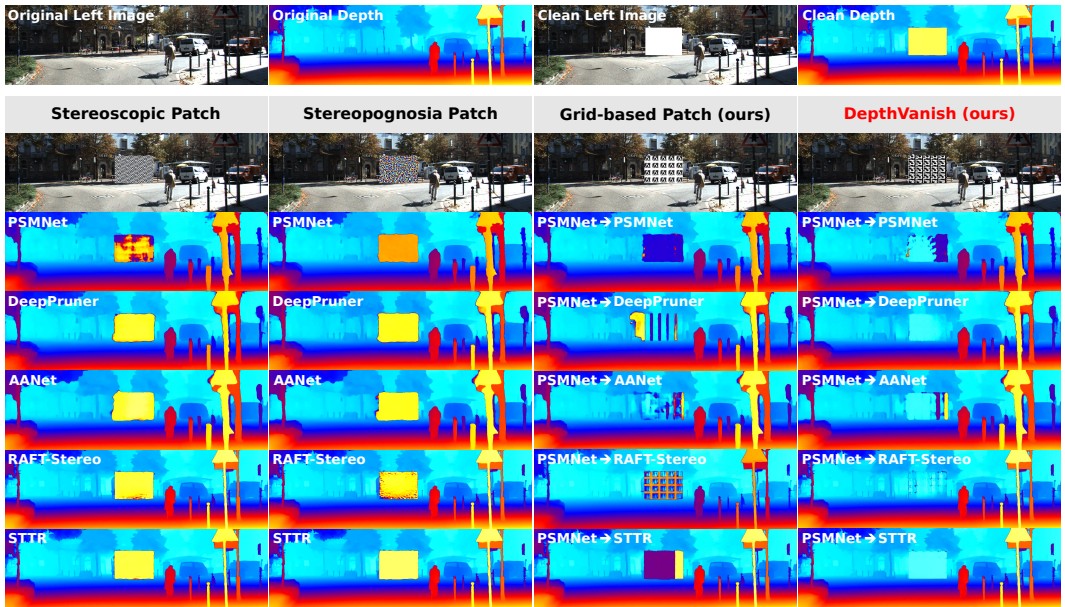

**Figure 5:** Visualization of different digital patch attack baselines and our DepthVanish patch against different target models on KITTI-scene dataset. Note that the original and clean depth are estimated by RAFT-Stereo.

where the bad pixel is one that satisfy $|\mathbf{d}_{pred} - \mathbf{d}_{gt}| > \max(3, 0.05 \cdot \mathbf{d}_{gt})$. To evaluate patch attack effectiveness, we first follow Eq. (4) to set the ground-truth disparity of the patch as $\mathbf{d}_{gt} = \mathbf{c}$. Then, we define the bad pixels as those satisfying $|\mathbf{d}_{pred} - \mathbf{c}| > \max(3, 0.05 \cdot \mathbf{d}_{gt})$ and $|\mathbf{d}_{pred} - \mathbf{0}| < \frac{\mathbf{c}}{n}$, where $n$ defines how many times deeper than the actual depth will a patch be considered to be attack effective. In summary, we report the average D1-error and EPE with standard deviation where higher values indicate better attack performance.

## 6.2 Digitally Attack Stereo Estimator

We first conduct digital attack experiments with our proposed DepthVanish patch on KITTI-scene dataset and the four sub-sets of DrivingStereo, *i.e.*, sunny, foggy, rainy, cloudy.

*Setting:* Due to the lack of existing works on attacking stereo matching using patches, we use the results from existing digital attack studies (*i.e.*, Stereoscopic [1] and Stereopagnosia [31]) as patches and deploy them into the scene as the first set of baselines. However, it should be noted that such comparison is not fair enough as existing works [1, 31] are not specifically designed for patch attack. Thus we further setup our own baseline (*i.e.*, grid-based patch from Sec. 5.1) for a fair comparison.

*Results:* ❶ We report the attack results for the five attack target models on KITTI-scene dataset in Tab. 1. It can be seen that existing digital attacks are ineffective under the patch attack setup, while our Grid-based Patch significantly outperforms them. Notably, DepthVanish achieves strong attack performance, especially against DeepPruner, RAFT-Stereo and STTR. We illustrate the results on DrivingStereo dataset in Fig. 4. It is evident that similar attack performance can be observed on all four sub-sets. ❷ In addition to the standard evaluation, Fig. 5 shows a KITTI sample comparison. Compared to the Clean Depth, we first note that existing attack works fail to mislead all the five target models, where only the Stereoscopic Patch shows limited influence against PSMNet. However, as the results shown in the last column, our DepthVanish patch casts strong influence where it almost disappeared within the depth results. More surprisingly, our patch enjoys significant transferability over models where the patch optimized with PSMNet shows strong attack effect on other four models. Based our experimental experience, all patches with such clear interval patterns are transferable across models, a capability we attribute to the insights analyzed in Sec. 3. Please refer to the supplement for the comprehensive experimental results of attack transferability.

## 6.3 Physically Attack Stereo Estimator

In this section, we conduct physical evaluation for our DepthVanish patches that optimized with different stereo estimators to highlight the importance and emergency of research on stereo matching

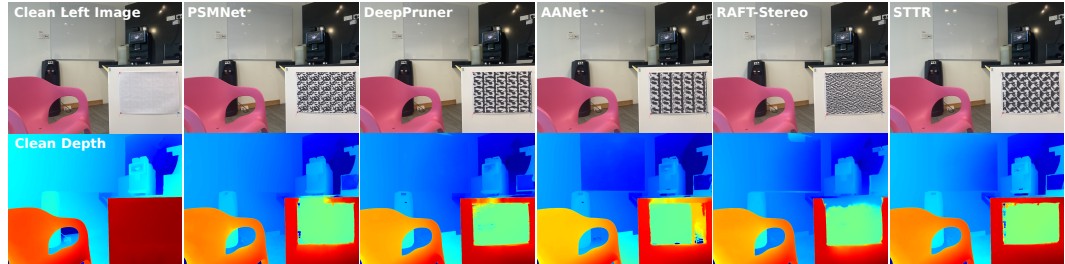

**Figure 6:** Visualization of physical attack results of our DepthVanish patches against different stereo depth estimators. Note that the clean depth is estimated with RAFT-Stereo.

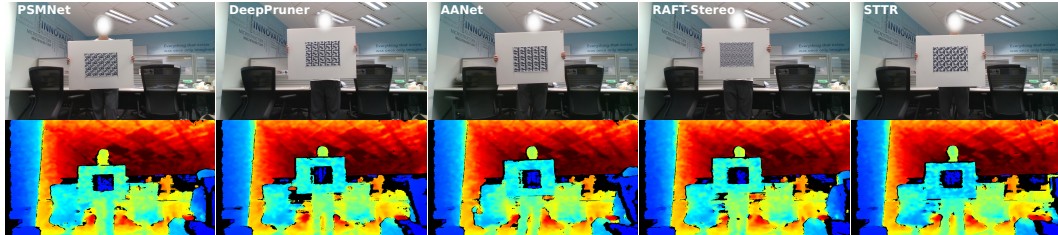

**Figure 7:** Visualization of DepthVanish attack performance against Intel RealSense depth camera (D435i).

reliability. As shown in Fig. 6, we host our DepthVanish patches on a white board for the purpose of highlighting the depth inconsistency. ❶ It can be observed from the results that our DepthVanish patch consistently preserves its attack effectiveness after deployed into the physical environment. Compared to the Clean Depth, the board region occupied by our DepthVanish patches are predicted as far away in general. ❷ However, it should be noted that the induced depth error are limited compared to the digital effectiveness in Fig. 5. We ascribe such performance degradation to the lighting variation and imprecise photo-capturing process, where the left and right images are captured manually and separately. Therefore, we further conduct evaluation for our patch against a commercial stereo depth camera in the next section. In summary, despite of the imprecise stereo image capturing process, our DepthVanish patches successfully attack advanced DNN-based stereo estimators with consistency.

## 6.4 Attack Commercial Stereo System

To further assess the practicality and robustness of our DepthVanish patch, we evaluate its performance on a commercial stereo camera system, specifically the Intel RealSense D435i depth camera. We focus on evaluating the patch's robustness from three aspects: model generalization, viewing orientation, and distance variation. ❶ **Model generalization:** we deploy the patches that optimized with five stereo models over KITTI dataset and evaluate their attack effective against D435i camera. As shown in Fig. 7, the patch consistently disrupts D435i predictions regardless of which model is optimized for, demonstrating strong attack transferability. ❷ **Orientation robustness:** we physically rotate the patch along the X and Y axes (see Fig. 3(a)). As visualized in Fig. 8, the patch (optimized with PSMNet on KITTI) remains effective under different viewing angles, confirming its robustness to rotation. ❸ **Distance robustness:** our method also shows robustness under varying distances. Corresponding visual results are provided in the supplementary material.

## 6.5 Ablation Study

In this section, we conduct ablation analysis on the DepthVanish attack to assess the impact of the hyperparameters $\alpha$ and $\beta$ in the objective function of Eq. (9). As shown in Fig. 9, both parameters are critical for optimal attack performance. Specifically, it can be seen that the performance degraded significantly when $\alpha = 0$, *i.e.*, the $\mathcal{L}_{entropy}$ is removed from Eq. (9), which highlights the importance

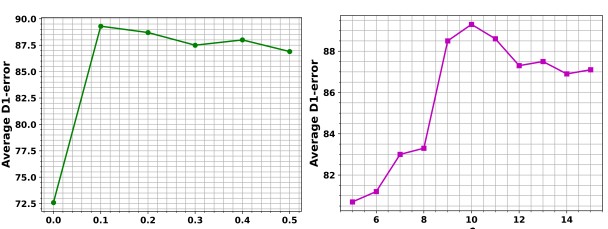

**Figure 9:** Attack performance of DepthVanish against RAFT-Stereo under different $\alpha$ and $\beta$ on KITTI dataset.

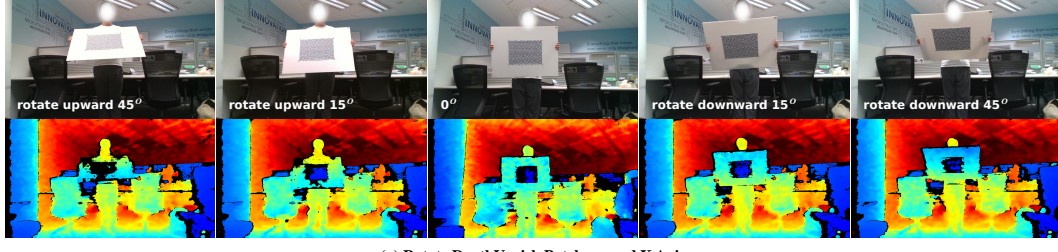

(a) Rotate DepthVanish Patch around X Axis

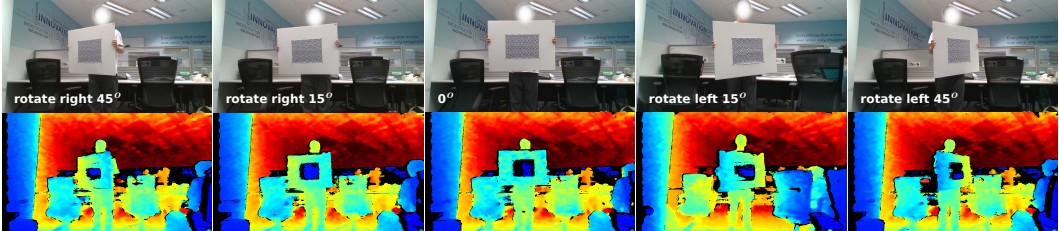

(b) Rotate DepthVanish Patch around Y Axis

**Figure 8:** Visualization of DepthVanish attack performance with different rotation degrees around both X and Y axes against Intel RealSense depth camera (D435i).

of the clear interval spacing for attack effectiveness. Moreover, the total variation constraint $\mathcal{L}_{tv}$ is also important where a clear performance degradation can be observed when $\beta$ decreases below 9. In summary, the synergistic combination of entropy and total variation regularization effectively ensures that our DepthVanish patches achieve the maximal attack performance

# 7  Conclusion

In this work, we present DepthVanish, a significant advancement in physical adversarial attack that jointly optimizes both texture element and interval structure of a patch to fool stereo depth estimation systems. By thoroughly analyzing the influence of regular spacing on naive texture repetition, we introduce a novel insight into enhancing the attack effectiveness and digital-to-physical transferability of the patch. To demonstrate the potentially dangerous consequences of depth estimation failure, we design the patch to be "disappear", where the patch is estimated as far away despite being physically close. Unlike previous methods limited to digital environments, our approach succeeds in both digital and physical settings, when evaluated against widely applied depth estimation models and commercial RGB-D cameras. These findings reveal critical vulnerabilities in current depth estimation technologies and raise concerns about their reliability in safety-critical autonomous systems.

## Acknowledgments and Disclosure of Funding

This research was supported by Shenzhen Science and Technology Program (No. JCYJ20240813114237048), "Science and Technology Yongjiang 2035" key technology breakthrough plan project (No. 2025Z053). This research is supported by the National Research Foundation, Singapore under its AI Singapore Programme (AISG Award No: AISG4-GC-2023-008-1B), and National Research Foundation, Singapore and Infocomm Media Development Authority under its Trust Tech Funding Initiative. Any opinions, findings and conclusions or recommendations expressed in this material are those of the author(s) and do not reflect the views of National Research Foundation, Singapore, Cyber Security Agency of Singapore, and Infocomm Media Development Authority. This work is also supported in part by Canada CIFAR AI Chairs Program, the Natural Sciences and Engineering Research Council of Canada, and JST-Mirai Program Grant No.JPMJMI20B8, JSPS KAKENHI Grant No.JP21H04877, No.JP23H03372, No.JP24K02920.

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
