# OpenReview forum: "DepthVanish: Optimizing Adversarial Interval Structures for Stereo-Depth-Invisible Patches"
_NeurIPS.cc/2025/Conference — NeurIPS 2025 poster_

### Official Review · Reviewer_xpZm · 2025-07-03

**Clarity:** 3
**Significance:** 3
**Originality:** 2
**Rating:** 4
**Confidence:** 4

**Summary:**

Stereo depth estimation is commonly used in safety-critical applications such as autonomous driving and robotics for depth and distance estimation. Various deep learning models have been proposed for highly accurate and robust depth estimation using stereo images. However, the literature has shown that these models are vulnerable to adversarial attacks. Most state-of-the-art works on adversarial attacks for depth estimation models have focused either on monocular depth estimation or solely on the digital domain. This work focuses on improving the effectiveness of adversarial attacks on stereo depth estimation models in the physical domain. The authors show that introducing regular intervals between repeated texture patterns can significantly enhance the effectiveness of patch attacks on stereo depth estimation models in the physical world. Based on these insights, the authors propose a novel attack, DepthVanish, that optimizes both the texture pattern and the interval (striped structure). The results highlight its effectiveness even on commercial Intel RealSense cameras.

**Questions:**

• How does the proposed technique perform under variations in ground truth depth and patch orientations along the x, y, and z axes in both digital and physical settings? Including plots similar to those in Fig. 2 and Fig. 3 for the evaluations in the results section can improve the understanding of the performance of the attack.

• What specific procedure was used to generate the patch shown in Fig. 2? How consistent are the results across different runs if the output of the procedure is not the same across runs?

• What is meant by “we empirically apply the optimal interval width o and repetition times k from Sec. 3”? How exactly one determines optimal interval width o and repetition times k?

• What is meant by “we keep adopt a patch with a physical size (u, v) = (0.891m, 1.26m) and specify the physical ground-truth depth e = 5m”? Why specifically just these values?

• The current discussion of related work seems limited, with comparisons made to only two prior works. To effectively position the proposed approach within the existing works, it is important to consider additional relevant references. Works that should be considered are:
A. Liu, et al. 2025. 'Optimization-Free Patch Attack on Stereo Depth Estimation.' arXiv preprint arXiv:2506.17632.
B. Wang, et al. 2024. 'Left-Right Discrepancy for Adversarial Attack on Stereo Networks.' arXiv preprint arXiv:2401.07188.
C. Liu, et al. 2024. 'Physical Attack for Stereo Matching.' In Proceedings of the International Conference on Computer Vision and Deep Learning (CVDL 2024)

**Ethical Concerns:**

["NO or VERY MINOR ethics concerns only"]

**Final Justification:**

I appreciate authors' efforts in providing additional discussion points, justifications, and results. Most of my major comments are addressed. It would be good to add figures in the revised manuscript in case accepted.
One comment, instead of cropping the patch from the paper Ref 3 because code is not available, at least try to implement the approach based on the methodology explained in the paper.
Anyway, I have upgraded my rating to Borderline Accept.

**Limitations:**

Yes

**Quality:**

3

**Strengths And Weaknesses:**

Strengths

• The paper presents a physically realizable adversarial attack on stereo depth estimation models. The attack is transferable across models, and demonstrated to work against even commercial solutions such as Intel RealSense.

• The attack is evaluated in both digital and physical domains, with multiple models and datasets used to demonstrate its effectiveness.


Weaknesses

• The paper provides limited coverage of related literature, focusing on only two prior works for comparison. To strengthen the contribution, additional relevant references should be considered and discussed.

A. Liu et al. 2025. “Optimization-Free Patch Attack on Stereo Depth Estimation.” arXiv preprint arXiv:2506.17632.

B. Wang et al. 2024. “Left-Right Discrepancy for Adversarial Attack on Stereo Networks.” arXiv preprint arXiv:2401.07188.

C. Liu et al. 2024. “Physical Attack for Stereo Matching.” In Proceedings of the International Conference on Computer Vision and Deep Learning (CVDL 2024)


• The empirical selection of some of the parameters is not well explained.

• Limited evaluation of the proposed technique. Important variations such as changes in physical ground truth depth, patch orientation across the x, y, and z axes, and lighting conditions are not considered. Additionally, the results section lacks plots, such as those in Fig. 2 and Fig. 3, that would illustrate the technique’s effectiveness in practical settings.

---

> ### Author Rebuttal · Authors · 2025-07-29
>
> ## 1. More Baselines.
> **W1: The paper provides limited coverage of related literature, focusing on only two prior works for comparison. To strengthen the contribution, additional relevant references should be considered and discussed.**
>
> **Q5: The current discussion of related work seems limited, with comparisons made to only two prior works. To effectively position the proposed approach within the existing works, it is important to consider additional relevant references.**
>
> Thank you for your valuable comments and for providing meaningful attack methods. We would like to note that we are familiar with the three related works listed. However, Reference 1 was submitted to arXiv one month after our NIPS submission. Reference 2 is similar to Stereopognosia (one of the baselines we compared against); both apply adversarial noise to the entire image for the left and right views, respectively. Regarding Reference 3, since the code has not been released, we directly cropped the relevant patch from the Reference 3 paper and used it for evaluation. The attack results for RAFT-Stereo and STTR on the KITTI dataset are presented in Tab. 4.1. Please note that the experimental setup remains the same as in our main experiments. Observing the results for new baselines, we see that our method consistently achieves better performance.
>
> Table 4.1: The attack performance of Reference 2 and Reference 3 on the KITTI dataset.
>
> | Method         | RAFT-Stereo D1      | RAFT-Stereo EPE     | STTR D1           | STTR EPE          |
> |----------------|----------------------|-----------------------|--------------------|--------------------|
> | Stereopagnosia | 4.18±11.27          | 2.09±12.66           | 3.02±3.00         | 1.28±8.79         |
> | Ref. 2           | 5.02±6.79           | 2.50±7.30            | 3.36±2.84         | 1.31±5.06         |
> |Ref. 3           | 5.88±2.30 | 3.24±6.90 | 3.04±9.17 | 1.69±10.27 |
> | Ours           | 89.31±6.56          | 66.01±6.18           | 92.38±8.76        | 69.25±6.62        |
>
> ## 2. Hyper-parameter Explanation
> **W2: The empirical selection of some of the parameters is not well explained.**
>
> **Q3: What is meant by “we empirically apply the optimal interval width o and repetition times k from Sec. 3”? How exactly one determines optimal interval width o and repetition times k?**
>
> Thank you for the comments. This means that the values for $o$ (optimal interval width) and $k$ (repetition times) are derived from the experimental investigations described earlier in the paper (specifically in Section 3). We further supplemented the experiment as follows:
>
> 1. To determine the optimal interval width $o$, we systematically studied the influence of different interval widths. Tab. 4.2 summarizes the attack performance of different interval width that averaged over various patch physical dpeth. Based on this investigation, 10 pixels yielded the best performance; therefore, $o = 10px$ is used as the optimal setting in the main experiments.
>
> 2. The number of repetitions $k$ is determined by the texture element $E$, which has a fixed size of $64 \times 64$ pixels, consistent with the *Stereoscopic* setting. $k$ is calculated based on how many such elements can fit within the patch using the chosen interval $o$, ensuring a dense yet non-overlapping arrangement.
>
> Table 4.2 The average attack performance for different interval widths on the KITTI scene.
>
> |  | 1px | 2px | 3px | 4px | 5px | 6px | 7px | 8px | 9px | 10px |
> |--|--|--|--|--|--|--|--|--|--|--|
> | D1-error | 17.76 | 20.14 | 19.09 | 20.72 | 23.22 | 22.57 | 23.15 | 22.99 | 22.90 | 25.44 |
>
> **Q4:  What is meant by “we keep adopt a patch with a physical size (u, v) = (0.891m, 1.26m) and specify the physical ground-truth depth e = 5m”? Why specifically just these values?**
>
> Given the objective of physical deployment, the patch size ratio is defined as $1:\sqrt{2}$, corresponding to the aspect ratio of A3 paper, to facilitate practical printing. To ensure methodological consistency, this configuration is also adopted during digital evaluation. In Section C.1 of the Supplemental Material, we ablated both of these parameters, and some combinations achieved better attack performance, e.g., $(u,v) = 0.8 \times (0.891m, 1.26m)$ with $e = 13m$. The main experiments use the combination $(u,v) = (0.891m, 1.26m)$ and $e = 5m$ for a clear visualization, as smaller patches or larger depths hinder visibility.
>
> ## 3. Experimental Details
> **Q2a: What specific procedure was used to generate the patch shown in Fig. 2?**
>
> The patch for the empirical study is constructed following the same procedure of grid-based attack described in line 178-184 of Sec. 5.1, and only the interval setting (width and insertion direction, either horizontal or vertical) of the patch is varied. Note that the overall content is centerted, which means that the grid of repeated texture elements $E$ is symmetrically placed within the patch $P$, with equal or nearly equal margins on all sides.
>
> **Q2b: How consistent are the results across different runs if the output of the procedure is not the same across runs?**
>
> The only two factors that will influence the empircal study are the scene image and patch physical size adopted. As we cannot upload figures for visualization, we note that the overall trending are almost the same as Fig.2(a) for various scene images and patch physical size during our comprehensive empirical investigation.
>
> ## 4. Comprehensive Ablation Results
> **W3: Limited evaluation of the proposed technique. Important variations such as changes in physical ground truth depth, patch orientation across the x, y, and z axes, and lighting conditions are not considered. Additionally, the results section lacks plots, such as those in Fig. 2 and Fig. 3, that would illustrate the technique’s effectiveness in practical settings.**
>
> **Q1: How does the proposed technique perform under variations in ground truth depth and patch orientations along the x, y, and z axes in both digital and physical settings? Including plots similar to those in Fig. 2 and Fig. 3 for the evaluations in the results section can improve the understanding of the performance of the attack.**
>
> **Physical Ground Truth Depth.** The digital and physical evaluation results of various physical ground truth depth are presented as Fig.2 and Fig.5 in the Supplemental Aaterial respectively. These evaluations demonstrate the robustness of the attack across a range of depth values.
>
> **Patch Orientation (Rotation).** While the impact of physical patch rotation is visualized in Fig. 8, we also provide digital evaluation results under different rotational axes in Table 4.3 using RAFT-Stereo on the KITTI dataset. Rotation along the Z axis falls outside the scope of our attack design, as it disrupts the necessary repetition along epipolar lines. Consequently, current attack methods (including Stereoscopic) are also unable to handle this type of rotation. Apart from this, our method demonstrates robustness to rotations along the X and Y axes.
>
> Table 4.3: The average attack performance (D1-error) under various rotation degree.
>
> | Axis | $0^o$ | $10^o$ | $20^o$ | $30^o$ | $40^o$ | $50^o$ | $60^o$ |
> |---|---|---|---|---|---|---|---|
> | X (ours) | 89.31±6.56 | 89.95±6.60 | 89.14±5.43 | 89.74±3.81 | 88.02±3.10 | 87.36±4.70 | 85.23±1.70 |
> | Y (ours) | 89.31±6.56 | 88.53±2.31 | 89.41±6.22 | 87.58±3.35 | 85.66±2.15 | 79.09±3.67 | 75.56±2.73 |
> ||
> | Z (ours) | 89.31±6.56 | 2.15±0.88 | 3.21±2.39 | 1.17±0.30 | 1.15±0.62 | 2.27±1.10 |2.87±1.93 |
> | Z (Stereoscopic) | 5.79±9.88 | 3.58±1.50 | 2.23±1.93 | 1.52±1.77 | 2.56±0.25 | 2.62±2.16 | 3.87±1.79 |
>
> **Lighting and Shadow Variations.** To evaluate the influence of environmental factors such as lighting and shadow on attack performance, we conduct experiments using RAFT-Stereo on the KITTI dataset under six levels of perturbation.
> Lighting variations are simulated by adjusting image brightness using the OpenCV example code: `np.clip(image * ratio, 0, 255)`, where a smaller ratio represents lower lighting intensity. Shadow variations are introduced by overlaying a semi-transparent black mask (RGBA: \[0, 0, 0, 127.5]) over the image patch, with the shadow progressively covering the patch from left to right based on the specified ratio.
> As shown in Table 4.4, our method demonstrates strong robustness under varying lighting and shadow conditions.
>
> Table 4.4: The attack performance (D1-error) under different lighting and shadow conditions.
>
> | Perturbation | $10\%$ | $20\%$ | $40\%$ | $60\%$ | $80\%$ | $100\%$ |
> |---|---|---|---|---|---|---|
> | Lighting | 88.64±7.81 | 89.87±7.62 | 89.64±6.20 | 88.80±6.65 | 88.43±7.62 | 89.31±6.56 |
> | Shdow   | 87.30±5.19 | 87.26±6.24 | 82.71±6.18 | 88.69±5.60 | 89.26±6.22 | 89.62±6.36 |
>
>
> Since the rebuttal does not support figures, we will update the manuscript later to incorporate all the comprehensive ablation results as figures to facilitate the intuitive understanding of our work.

---

> ### Author Response · Authors · 2025-08-05
>
> Dear Reviewer xpZm,
>
> We would like to kindly remind you that the discussion period is nearing its end. Your feedback is extremely valuable to us, and we remain fully open and eager to further clarify or improve our work based on your insights. Please don’t hesitate to let us know if you have any further concerns or suggestions.
>
> Best regards,
>
> The Authors of DepthVanish

---

> ### Comment · Reviewer_xpZm · 2025-08-05
> **Responses and additional results are good**
>
> I appreciate authors' efforts in providing additional discussion points, justifications, and results. Most of my major comments are addressed. It would be good to add figures in the revised manuscript in case accepted.
>
> One comment, instead of cropping the patch from the paper Ref 3 because code is not available, at least try to implement the approach based on the methodology explained in the paper.
>
> Anyway, I will upgrade my rating to Borderline Accept.

---

> > ### Author Response · Authors · 2025-08-06
> >
> > Thank you very much for raising the rating of our submission. To further improve the quality of the manuscript, we will incorporate the visualizations corresponding to the additional experimental results in the revised version.
> >
> > Regarding the evaluation of Ref. 3, we appreciate your suggestion. In the absence of publicly available code, we have reimplemented the approach based on the methodology described in the paper, and we summarize the results in Tab. 4a below. We observe that the patches from reimplemented Ref. 3 are still unable to effectively attack RAFT-Stereo and STTR.
> >
> > Table 4a: The attack performancee of reimplemented Reference 3 on KITTI dataset.
> >
> > | Method         | RAFT-Stereo D1      | RAFT-Stereo EPE     | STTR D1           | STTR EPE          |
> > |----------------|----------------------|-----------------------|--------------------|--------------------|
> > |Ref. 3           | 6.90±4.79 | 2.37±3.81 | 5.46±2.76 | 3.02±1.79 |
> > | Ours           | 89.31±6.56          | 66.01±6.18           | 92.38±8.76        | 69.25±6.62        |
> >
> > We greatly appreciate your thoughtful feedback and the opportunity to address the further concern. We hope the additional experiments resolve the concern, and please do not hesitate to let us know if there are any further issues regarding the manuscript.
> >
> > Best Regards

---

> > > ### Comment · Reviewer_xpZm · 2025-08-06
> > > **Excellent effort in the rebuttal**
> > >
> > > Thanks for providing extra results and considering the feedback positively. I don't have any further comments.

---

> > > > ### Author Response · Authors · 2025-08-07
> > > >
> > > > Thank you for your time and constructive feedback throughout the discussion. We're glad the additional results addressed your concerns, and we truly appreciate your support.
> > > >
> > > > Best Regards

---

### Official Review · Reviewer_aSEK · 2025-07-03

**Clarity:** 3
**Significance:** 2
**Originality:** 2
**Rating:** 4
**Confidence:** 4

**Summary:**

This paper focus on Adversarial attacks against stereo depth estimation and shows that inserting regular gaps to form striped patterns, rather than naively repeating textures, which dramatically improves the real-world effectiveness of adversarial patches against stereo depth estimation. By jointly optimizing stripe intervals and texture content, these patches reliably fool state-of-the-art stereo models (RAFT-Stereo, STTR) and even commercial Intel RealSense RGB-D cameras under real-world conditions.

**Questions:**

1. The authors are suggested to provide a more intuitive example to illustrate why the proposed attack is effective. For instance, in transformer-based methods, could you present depth attention maps showing the value changes before and after the attack to draw further interesting conclusions?
2. More stereo depth estimation methods should be included to validate the attack’s effectiveness, preferably on the recently robust 3D foundation model VGGT.
3. Could the authors provide further theoretical analysis or insights into the inductive biases of the proposed method?

**Ethical Concerns:**

["NO or VERY MINOR ethics concerns only"]

**Final Justification:**

Thanks for the response from the authors. The authors conducted more experiments on four stereo estimators and the VGGT backbone. The added results demonstrate the strong performance, showing the generalization ability of the proposed method. Since the authors addressed my concerns well, I would like to upgrade my rating to positive. The authors are also encouraged to fix other issues, like low image resolution.

**Limitations:**

Yes

**Paper Formatting Concerns:**

No paper formatting concerns

**Quality:**

3

**Strengths And Weaknesses:**

Strengths:
1. This paper introduces the first adversarial patch that reliably transfers from digital simulation to real-world stereo setups, encompassing both learning-based (RAFT-Stereo, STTR) and commercial (Intel RealSense) systems.
2. Through systematic empirical analysis, the authors identify that inserting regular intervals between repeated textures (striped patterns) dramatically improves physical-world attack transferability, a key observation not previously reported.

Weaknesses:
1. The stereo matching methods targeted in this paper are not the latest in the field; for example, the authors do not validate their proposed attack on classic approaches such as [1], [2], and [3].
2. The image clarity in this paper has some issues, for example, in Fig. 2(a) and Figs. 3(b) and 3(c).
3. There are some typos; for example, on line 16, “RAFTS-tereo” should be corrected to “RAFT-Stereo.”
4. This paper should contain further theoretical analysis or insights into the proposed method.

[1] Iterative Geometry Encoding Volume for Stereo Matching (Xu et al., CVPR 2023)

[2] Practical Stereo Matching via Cascaded Recurrent Network With Adaptive Correlation (Li et al., CVPR 2022)

[3] Unifying Flow, Stereo, and Depth Estimation.(Xu et al., TPAMI 2023)

---

> ### Author Rebuttal · Authors · 2025-07-31
>
> ## 1. More Attack Targets
> **W1: The stereo matching methods targeted in this paper are not the latest in the field; for example, the authors do not validate their proposed attack on classic approaches such as [1], [2], and [3].**
>
> Thank you for the suggestion. As a supplement, we further evaluate our method by attacking four stereo estimators: GMFlow-unify [3], IGEV-Stereo [1], ACVNet [5], and CREStereo [2], using the same setup as in our main experiments. As shown in Tab. 3.1, our method consistently demonstrates strong attack performance on new target models.
>
> Table 3.1: Attack performance on four additional target models using the KITTI dataset.
>
> |  | GMFlow-unify D1 | GMFlow-unify EPE | IGEV-Stereo D1 | IGEV-Stereo EPE | ACVNet D1 | ACVNet EPE | CREStereo D1 | CREStereo EPE |
> |--|--|--|--|--|--|--|--|--|
> | Stereoscopic | 5.39±7.34 | 3.89±6.40 | 6.66±7.88 | 4.32±5.73 | 2.96±2.08 | 1.67±3.77 | 5.72±6.25 | 4.62±6.28 |
> | ours | 86.65±3.19 | 68.61±6.10 | 87.53±4.15 | 66.24±5.92 | 69.16±4.92 | 52.27±8.10 | 86.03±5.35 | 64.59±6.26 |
>
> **Q2: More stereo depth estimation methods should be included to validate the attack's effectiveness, preferably on the recently robust 3D foundation model VGGT.**
>
> Thank you for the valuable suggestion. To further validate the effectiveness of our attack, we evaluate it on the recently introduced robust 3D foundation model VGGT [4]. To facilitate this evaluation, we directly use our patch optimized with RAFT-Stereo and the Stereoscopic patch, and evaluate them on the DTU dataset [6]. As shown by Tab. 3.2, our attack still achieves much higher D1 and EPE errors compared to the baseline, thereby demonstrating its effectiveness even against a state-of-the-art robust model.
>
> Table 3.2: Attack performance on VGGT using the DTU dataset.
>
> |Method| RAFT-Stereo D1 | RAFT-Stereo EPE |
> |--|--|--|
> | Stereoscopic | 6.96±1.08 | 5.67±0.77 |
> | Ours | 42.29±8.90 | 37.84±5.10 |
>
> ## 2. Figure Resolution
> **W2: The image clarity in this paper has some issues, for example, in Fig. 2(a) and Figs. 3(b) and 3(c).**
>
> Thank you for the feedback. The reduced clarity of the figures is due to the compression applied during manuscript submission. We will provide higher-resolution versions of all figures in the revised manuscript to ensure they are clearly visible and easy to read.
>
> ## 3. Writing & Typos Issues
> **W3: There are some typos; for example, on line 16, “RAFTS-tereo” should be corrected to “RAFT-Stereo.”**
>
> Thank you for the feedback. We will carefully revise all of the writing issues for the updated version.
>
> ## 4. Theoretical & Intuitive Analysis
>
> **W4: This paper should contain further theoretical analysis or insights into the proposed method.**
>
> **Q1: The authors are suggested to provide a more intuitive example to illustrate why the proposed attack is effective. For instance, in transformer-based methods, could you present depth attention maps showing the value changes before and after the attack to draw further interesting conclusions?**
>
> Thank you for the suggestion. We conducted a saliency map analysis [7] to identify which regions in the input image most influence the model's depth prediction for the patch. We compared two patch types under identical KITTI scene setups: (1) a naive repeated patch (Fig. 2(b), top-left), and (2) our proposed DepthVanish patch with interval insertion. While figures cannot be included in the rebuttal, we describe the results in words and will include the visualizations in the revised version.
>
> **Observations:**
> * For the **naive patch**, the model's prediction is heavily influenced by the surrounding scene content; saliency is broadly distributed.
> * For the **DepthVanish patch**, the model's focus shifts toward the patch region itself, indicating increased reliance on the patch content.
>
> **Conclusion:**
> The above observation provides us a potentional explanation of why interval inserting can attack more effectively. The introduction of intervals makes the depth prediction of the patch more focus on the patch region itself where the repeated pattern presents. As it is well-known that epipolar geometry becomes unrealiable when repeated pattern exists, our method enhances such unrealiability by shifting the model focus onto the patch region during prediction.
>
> **Q3: Could the authors provide further theoretical analysis or insights into the inductive biases of the proposed method?**
>
> We appreciate the insightful suggestion and agree that a deeper understanding of the inductive biases in our method is valuable. While our primary focus has been empirical, we offer the following analysis:
>
> 1. **Structured Ambiguity Bias.**
> Our method is built on the theoretical insight that stereo matching systems are fundamentally vulnerable to periodic patterns that create correspondence ambiguity (Eq. 2). Such ambiguity is amplified by grid intervals, which act as structured aliasing mechanisms creating extensive local minima in the cost volume. The inductive bias here is that spatial discontinuities (intervals) combined with repetitive patterns create stronger matching confusion than continuous repetitive patterns alone.
>
> 2. **Multi-directional Disruption Bias.**
> Our grid-based interval structure incorporates the bias that stereo systems are most vulnerable when correspondence ambiguity occurs in both horizontal (epipolar) and vertical directions simultaneously. This reflects the underlying assumption that stereo algorithms rely on both local feature matching and global consistency constraints, disrupting both dimensions maximizes attack effectiveness.
>
> 4. **Binary Structure Regularization.**
> Our entropy and total variation losses, i.e., Eq. (7) and Eq. (8), encode the inductive bias that sharp, well-defined boundaries between texture and interval regions are more effective than gradual transitions. This reflects the insight that stereo algorithms are more confused by clear structural patterns than by smooth variations, especially for the physical scenarios.
>
> References
>
> [1] Iterative Geometry Encoding Volume for Stereo Matching (Xu et al., CVPR 2023)
>
> [2] Practical Stereo Matching via Cascaded Recurrent Network With Adaptive Correlation (Li et al., CVPR 2022)
>
> [3] Unifying Flow, Stereo, and Depth Estimation.(Xu et al., TPAMI 2023)
>
> [4] Vggt: Visual geometry grounded transformer. (Wang et al., CVPR 2025)
>
> [5] Attention concatenation volume for accurate and efficient stereo matching. (Xu, Gangwei, et al. CVPR 2022)
>
> [6] Large scale multi-view stereopsis evaluation. (CVPR 2014)
>
> [7]  Deep inside convolutional networks: Visualising image classification models and saliency maps. (Simonyan K, et al., arXiv)

---

> ### Author Response · Authors · 2025-08-05
>
> Dear Reviewer aSEK,
>
> We would like to kindly remind you that the discussion period is nearing its end. Your feedback is extremely valuable to us, and we remain fully open and eager to further clarify or improve our work based on your insights. Please don’t hesitate to let us know if you have any further concerns or suggestions.
>
> Best regards,
>
> The Authors of DepthVanish

---

> ### Author Response · Authors · 2025-08-08
> **Gentle Reminder: NIPS Discussion Period Closing Soon**
>
> Dear Reviewer aSEK,
>
> We hope this message finds you well.
>
> As the discussion period will conclude in less than 36 hours, we would like to respectfully follow up on our earlier message and kindly invite you to share any additional thoughts or concerns regarding our submission.
>
> Your insights are highly valued, and your input would greatly contribute to a balanced and thorough evaluation. If there are any remaining issues or questions, we would be more than happy to provide further clarification.
>
> Thank you again for your time and consideration.
>
> Best regards,
>
> The Authors of DepthVanish

---

### Official Review · Reviewer_FaQu · 2025-07-04

**Clarity:** 2
**Significance:** 3
**Originality:** 3
**Rating:** 4
**Confidence:** 3

**Summary:**

The paper presents "DepthVanish," an adversarial patch attack for stereo depth estimation that optimizes both texture and interval structures to enhance physical-world effectiveness. Previous work on repetitive textures failed in recent SOTA methods, but the authors discover that introducing regular intervals (striped/grid structures) significantly improves attack performance. Based on this findings, they develop a joint optimization pipeline to co-design texture elements and spatial layouts, enabling the patch to mislead state-of-the-art models (e.g., RAFT-Stereo, STTR) and commercial RGB-D cameras (Intel RealSense) in both digital and physical settings. Experiments show that structured intervals boost depth misestimation (e.g., predicting infinite depth for close objects) and withstand viewpoint rotations, highlighting critical vulnerabilities in stereo systems.

**Questions:**

1. Could the authors evaluate if the designed method can be applied to flow matching and related scenario? This can further demonstrate the generalizability of the designed attack method.

**Ethical Concerns:**

["NO or VERY MINOR ethics concerns only"]

**Final Justification:**

The author response and additional experimental results address most of my concern, hence I keep my positive rating.

**Limitations:**

yes

**Quality:**

3

**Strengths And Weaknesses:**

### Strengths
1. Effective attack designs. The authors introduce an interval-based structural design that demonstrates significant attack efficacy on stereo depth estimation systems across both digital simulations (e.g., KITTI, DrivingStereo) and physical deployments (e.g., Intel RealSense).

2. Joint Optimization pipeline. Building on the interval structure insight, the proposed pipeline co-optimizes texture elements and spatial layouts, enhancing attack transferability across diverse models (e.g., RAFT-Stereo, STTR) and environmental conditions.

3. Comprehensive evaluation. Experiments span digital evaluations on benchmark datasets and physical tests on commercial hardware, showcasing practical utility and real-world relevance for safety-critical systems.


### Weakness
1. Writing issue. The figures in the paper (e.g., depth prediction visualizations) lack sufficient resolution or labeling, potentially impeding reader comprehension of attack performance comparisons.


2. Limited applicability. The attack evaluation is currently confined to stereo matching tasks, and the authors do not explore generalizability to related domains (e.g., optical flow estimation or multi-view stereo), which could expand the method’s utility.

---

> ### Author Rebuttal · Authors · 2025-07-31
>
> ## 1. Writing issue.
> **W1: The figures in the paper (e.g., depth prediction visualizations) lack sufficient resolution or labeling, potentially impeding reader comprehension of attack performance comparisons.**
>
> Thank you for the feedback. The reduced clarity of the figures is due to the compression applied during manuscript submission. We will provide higher-resolution version of all figures in the revised manuscript to ensure that all values are clearly visible and easy to read.
>
> ## 2. Limited applicability.
> **W2: The attack evaluation is currently confined to stereo matching tasks, and the authors do not explore generalizability to related domains (e.g., optical flow estimation or multi-view stereo), which could expand the method’s utility.**
>
> We appreciate your insightful suggestion regarding the potential extension of our work. In response, we present further analysis and experimental results for both the *optical flow* and *multi-view stereo* tasks.
>
> ### Optical Flow
>
> Evaluating adversarial attacks on **scene flow estimation**, which integrates both optical flow and stereo depth, largely follows the same fundamental principles as attacking stereo depth estimation. Our primary objective remains consistent: to perturb the depth predictions obtained from paired stereo images.
> The key distinction in the scene flow setting lies in the evaluation process: performance is averaged across a sequence of depth predictions, meaning the adversarial patch is viewed under varying angles and scales across consecutive stereo frames.
>
> To assess the robustness of our patch under such dynamic conditions, we consider both **rotational** and **scaling** variations:
>
> * **Rotation:** We have already visualized the physical impact of patch rotation in Fig. 8. Additionally, Tab. 2.1 presents digital evaluation results under simulated rotations along the X and Y axes. These results show minimal performance degradation, indicating that the patch remains effective across a wide range of viewing angles.
>
> * **Scale:** We present additional results in Fig. 2 and Fig. 5 of the Supplemental Material, which further demonstrate that the patch retains high attack efficacy across different physical scales.
>
> In conclusion, these findings suggest that our framework provides a strong foundation for attacking depth estimation in more complex 3D motion settings. Given the robustness of our method to viewpoint and scale variations, we expect similar levels of effectiveness in the scene flow context.
>
>
> Table 2.1: Average attack performance (D1-error) under different rotation angles.
>
> | Axis | $0^o$ | $10^o$ | $20^o$ | $30^o$ | $40^o$ | $50^o$ | $60^o$ |
> |---|---|---|---|---|---|---|---|
> | X | 89.31±6.56 | 89.95±6.60 | 89.14±5.43 | 89.74±3.81 | 88.02±3.10 | 87.36±4.70 | 85.23±1.70 |
> | Y | 89.31±6.56 | 88.53±2.31 | 89.41±6.22 | 87.58±3.35 | 85.66±2.15 | 79.09±3.67 | 75.56±2.73 |
>
> ### Multi-View Stereo
>
> As an initial investigation into the multi-view stereo setting, we adopt the recent VGGT model  [1] as the target for evaluating our adversarial patch. To facilitate this evaluation, we directly use our patch optimized with RAFT-Stereo and the Stereoscopic patch, and evaluate them on the DTU dataset [2].
> The average results in Tab. 2.2 shows our method still performs significantly better than the baseline method.
>
> Table 2.2: Attack performance on VGGT using DTU scenes.
>
> |Method| D1 | EPE |
> |--|--|--|
> | Stereoscopic | 6.96±1.08 | 5.67±0.77 |
> | Ours | 42.29±8.90 | 37.84±5.10 |
>
>
> References:
>
> [1] Vggt: Visual geometry grounded transformer. (Wang et al., CVPR 2025)
>
> [2] Large scale multi-view stereopsis evaluation. (CVPR 2014)

---

> > ### Comment · Reviewer_FaQu · 2025-08-06
> > **Reply by Reviewer**
> >
> > Thanks for your response. The experimental results on the scene flow and multi-view stereo effectively enhance the generalization ability of the paper. I will keep my positive rating. Please also update the image clarity in the revision.

---

> > > ### Author Response · Authors · 2025-08-07
> > >
> > > Thank you for your kind feedback and continued support. We’re glad that the additional experiments helped clarify the generalization ability of our approach. We will make sure to improve the image clarity in the revision as suggested.
> > >
> > > Best Regards

---

> ### Author Response · Authors · 2025-08-05
>
> Dear Reviewer FaQu,
>
> We would like to kindly remind you that the discussion period is nearing its end. Your feedback is extremely valuable to us, and we remain fully open and eager to further clarify or improve our work based on your insights. Please don’t hesitate to let us know if you have any further concerns or suggestions.
>
> Best regards,
>
> The Authors of DepthVanish

---

### Official Review · Reviewer_ZGfn · 2025-07-15

**Clarity:** 1
**Significance:** 3
**Originality:** 3
**Rating:** 5
**Confidence:** 4

**Summary:**

This work proposes a new adversarially optimized targeted patch attack for stereo depth estimation using DNNs.
The proposed method, DepthVanish, optimizes the adversarial patches for structured interval spacing and texture.

The manuscript shows the effectiveness of the optimized patches both digitally and in physical settings, including transfer attacks by optimizing the patches for one architecture and transferring the patch as an attack to another architecture.

The visualizations and empirical results demonstrate the effectiveness of the optimized patches.

**Questions:**

One major question would be:
In the manuscript, it is not clear, both in the formulation and in the visualizations, whether the patch is merely added to one image (left) or to both images from stereo, and if it is both, then how does the method ensure consistency in the patch? Or is this something that is not of concern, and if so, then discussion around this would significantly help with clarity.

Addressing the weaknesses would help improve the clarity of the work and the quality of the manuscript.

**Ethical Concerns:**

["NO or VERY MINOR ethics concerns only"]

**Final Justification:**

The authors have addressed my concerns.

From what I see, they have also addressed the major concerns from the other reviewers (some concerns were overlapping).

The work is good, the results make sense, and would lead to interesting discussions at NeurIPS.

**Limitations:**

Yes, the limitations are discussed briefly in the supplementary material. It would be great to include those in the main paper.

**Paper Formatting Concerns:**

No paper formatting concerns.

**Quality:**

1

**Strengths And Weaknesses:**

Strengths:

1. Visualizations and empirical results demonstrate the effectiveness of the optimized patches.
2. This work addresses an important real-world issue of vulnerability of DNN-based stereo depth estimation methods to malicious agents.
3. The manuscript shows the effectiveness of the optimized patches both digitally and in physical settings.
4. The optimized patch attacks for one architecture transfer to another architecture quite effectively.

Weaknesses:
1. The manuscript needs significant improvement in writing to add clarity. For example, section 3 already discusses Vertical and Horizontal Intervals as methods; however, they are defined significantly later in section 5. In the manuscript, it is not clear, both in the formulation and in the visualizations, whether the patch is merely added to one image (left) or to both images from stereo, and if it is both, then how does the method ensure consistency in the patch? Or is this something that is not of concern, and if so, then discussion around this would significantly help with clarity.
2. The resolution of the included figures is very low, and improving that would help. For example, in Figure 4, it is sometimes very difficult to read some of the values on the spider plot.
3. The manuscript repeatedly acknowledged RAFT-Stereo as the SotA method; however, on various datasets, numerous other methods outperform RAFT-Stereo, one example being [1] on the Spring dataset.
4. The manuscript does not evaluate their attack against any defense methods, one such example would be [2].
5. The manuscript in its current state is poorly written. For example, the related work paragraph on stereo depth estimation and depth estimation attack appears to be LLM-generated. This assessment is not merely because of the repeated use of "—" as done by many LLMs, but also because these paragraphs do not really do the work of the related work section in a manuscript. The paragraph on stereo depth estimation is more fitting as an "introduction" paragraph, since the stereo estimation methods used in this work are not really discussed here. The related work section is an opportunity to discuss how the different stereo-depth estimation methods used in the manuscript work and differ from each other. Likewise, the "depth estimation attack" paragraph also goes off-topic. For example, it talks about the lack of "effective defense strategies"; however, this work neither proposes nor uses any "defense strategies". Another example of poor writing is a lack of consistency. Line 33 writes "..Fig. 1..", whereas Line 39 writes "Figure 1". It would be great if these inconsistencies could be fixed.
6. The formulations lack definitions, for example, in equations 1 and 2, "d" is not defined, and in equation 4, the spatial resolution of d^p_gt is not defined, which I infer from the visualizations is not equal to the spatial resolutions of I_l and I_r.
7. In the conclusion, the manuscript claims that the proposed method effectively "disappears" objects; however, from the visualizations, this is an overclaim. Objects do not seem to disappear in the shared visualizations, only the region with the patch, regions in which the patch is occluding the objects, and the model seems to predict 0 disparity.
8. The manuscript only explores one target, ie, disparity=0 and thus depth=infinity; it would be interesting to see if the methodology could optimize the patch for other targets, for example, disparity=infinity, ie, depth=0.
9. There are multiple typos in the current manuscript, for example, in the abstract, Line 16: RAFTS-tereo instead of RAFT-Stereo, and in Line 226: missing space between (1242,* *375) while noting the image size for KITTI.


References:

[1] Xu, Gangwei, et al. "Attention concatenation volume for accurate and efficient stereo matching." Proceedings of the IEEE/CVF conference on computer vision and pattern recognition. 2022.

[2] Agnihotri, Shashank, Julia Grabinski, and Margret Keuper. "Improving feature stability during upsampling–spectral artifacts and the importance of spatial context." European Conference on Computer Vision. Cham: Springer Nature Switzerland, 2024.

---

> ### Author Rebuttal · Authors · 2025-07-31
>
> ## 1. Misinterpretation of Experimental Setup.
> **W1b: In the manuscript, it is not clear, both in the ... Or is this something that is not of concern, and if so, then discussion around this would significantly help with clarity.**
>
> Thank you for the comments. We clarify that the patch is applied to both the left and right images, and we have emphasized this in the manuscript through the following two aspects:
>
> 1. **Physical Attack Evaluation**: One of our main contributions is demonstrating our patch is physically effective. During physical evaluation, the optimized patch is placed in real-world scenes, then stereo images of these scenes are captured. As a result, the patch naturally appears in both the left and right images after capture.
>
> 2. **Digital Deployment**: We explicitly described the digital deployment procedure in Sec. 6.1 and Supplementary Material B.1, ensuring the patch is inserted into both stereo images following physical constraints. This guarantees consistency between digital and physical settings, meaning the patch is presented in both the left and right images during digital evaluation as well.
>
> Additionally, we clarify this in the formulation. In line 151 of the manuscript, we specified that the patch is added to the paired stereo images $\hat{I}_l$ and $\hat{I}_r$, contrasting with the clean inputs ${I}_l$ and ${I}_r$. To improve clarity, we have updated Eq. (3) (see response to W6) to explicitly show that the patch is applied to both stereo images. For visualization, we follow the standard practice of displaying only the left image during qualitative evaluations, since the depth map is typically aligned with the left image content.
>
> **W7: In the conclusion, the manuscript claims that the proposed method effectively "disappears" objects ... the model seems to predict 0 disparity.**
>
> Thank you for the comments. The concern stems from a misunderstanding: we do not claim that the physical object to which the patch is attached disappears from the image. Instead, the objective of our paper is to optimize the patch so that the depth estimation network interprets both the patch and the occluded object as being at an extremely large distance, making them appear to "disappear" in the predicted depth map. This poses a serious safety risk, as an autonomous agent that perceives an object ahead as infinitely far away may fail to brake and continue moving forward.
>
> **W8: The manuscript only explores one target, ie, disparity=0 ... other targets, for example, disparity=infinity, ie, depth=0.**
>
> Thank you for the insightful suggestion. We intentionally target depths approaching infinity (disparity near zero) because making an obstacle “vanish” poses a uniquely dangerous failure mode. An autonomous agent perceiving an object as infinitely far may continue moving without braking, potentially causing catastrophic consequences. This aligns with prior depth estimation attacks [3][7], reflecting a common adversarial goal with serious safety implications. In contrast, optimizing for disparity=infinity (very close depth) is generally less critical, where mistakenly detecting a distant object as close usually results in braking that is a safer outcome.
>
> As a complementary experiment, we optimize our patch toward disparity = infinity. Using the same patch setup as in the main experiments, we evaluate the average predicted depth as the ground-truth depth varies from $5m$ to $21m$. As shown in Table 1.1, the optimization still converges reliably.
>
> Table. 1.1: Average patch depth predictions on KITTI using RAFT-Stereo.
>
> |gt depth (m)|5|7|9|11|13|15|17|19|21|
> |-|-|-|-|-|-|-|-|-|-|
> |avg. predicted depth (m)|1.20±0.03|1.24±0.15|1.27±0.06|1.39±0.08|2.60±0.21|2.23±1.29|5.92±2.08|5.61±2.19|6.37±3.23|
>
>
> ## 2. Figure Resolution.
> **W2: The resolution of the included figures is very low, ... very difficult to read some of the values on the spider plot.**
>
> Thank you for your feedback. The reduced clarity of figures, including Figure 4, is due to the compression applied during manuscript submission. We will provide higher-resolution version of all figures in the revised manuscript to ensure that all values, especially those in the spider plot, are clearly visible and easy to read.
>
> ## 3. More Attack Targets.
> **W3: The manuscript repeatedly acknowledged RAFT-Stereo as the SotA method; ... being [1] on the Spring dataset.**
>
> We kindly note that our attack targets real-world traffic scenarios, so we evaluate it on the KITTI and DrivingStereo datasets rather than the Spring dataset. We select RAFT-Stereo and STTR as baselines because they represent distinct and pioneering approaches to stereo matching, enabling us to assess performance across different estimation strategies.
>
> To further demonstrate generalizability, we evaluate our method on four additional models: GMFlow-unify [6], IGEV-Stereo [4], ACVNet [1], and CREStereo [5], using the same experimental setup described in our paper. As shown in Tab. 1.2, our method consistently achieves successful attacks across all target models.
>
> Table 1.2: Attack performance on four additional target models using the KITTI dataset.
>
> |  |GMFlow-unify D1|GMFlow-unify EPE|IGEV-Stereo D1|IGEV-Stereo EPE|ACVNet D1|ACVNet EPE|CREStereo D1|CREStereo EPE|
> |-|-|-|-|-|-|-|-|-|
> |Stereoscopic|5.39±7.34|3.89±6.40|6.66±7.88|4.32±5.73|2.96±2.08|1.67±3.77|5.72±6.25|4.62±6.28|
> |Ours|86.65±3.19|68.61±6.10|87.53±4.15|66.24±5.92|69.16±4.92|52.27±8.10|86.03±5.35|64.59±6.26|
>
> ## 4. Possible Defenses.
> **W4: The manuscript does not evaluate their attack against any defense methods, one such example would be [2].**
>
> Thank you for the suggestion. To address this, we implemented the defense strategy from [2] by altering the kernel sizes in the transposed convolution layers used for pixel-wise upsampling in the “feature extractor” module of STTR from $3\times3$ to $7\times7$ + $3\times3$. As shown in Tab. 1.3, our attack remains effective under this modification.
>
> Table 1.3: Attack performance on STTR with modified kernel sizes.
>
> | |kernel size|D1|EPE|
> |-|-|-|-|
> |Without Defense|$3\times3$|92.38±8.76|69.25±6.62|
> |With Defense|$7\times7$+$3\times3$|90.71±8.08|69.67±5.77|
>
> ## 5. Writing Clarity
> **W1a: The manuscript needs significant improvement in writing ... they are defined significantly later in section 5.**
>
> **W5a: The manuscript in its current state is poorly written. ... however, this work neither proposes nor uses any "defense strategies".**
>
> Thank you for your comments and suggestions.
>
> 1. We agree that a clearer explanation of vertical and horizontal intervals is needed to clarify the patch construction procedure. Fig. 2(b) will be updated with a higher-resolution version for illustrative purpose, and the corresponding construction formulation will be added in the revised manuscript.
>
> 2. The use of LLMs was strictly limited to language refinement. All scientific content, experiments, and structure are our own. Stylistic issues (e.g., overuse of dashes) will be addressed.
>
> 3. We acknowledge that our discussion of stereo depth estimation can be expanded. The revised manuscript will include a more comprehensive review of the attack targets evaluated and recent related studies.
>
> 4. The “Depth estimation attack” paragraph is not off-topic. It traces the evolution from digital monocular attacks to physical stereo attacks to highlight the distinct challenges we address. Existing defenses are mentioned only to motivate the need for stronger, architecture-agnostic attacks as catalysts for future defense research.
>
> **W6: The formulations lack definitions, for example, ... not equal to the spatial resolutions of I_l and I_r.**
>
> Thank you for your feedback.
>
> * Regarding the variable $d$ in Eq. (1) and Eq. (2), line 85 clearly indicates that $d$ is the pixel offset relative to $x$. To make this definition more explicit, we will add the following immediately after the equations: "$d \in \mathbb{Z}$ represents the pixel-wise horizontal disparity between the left and right images."
>
> * Concerning the spatial resolution of $d^p_{gt}$, its size is the same as the patch size $(h_p, w_p)$. We will revise Eq. (3) and line 151 to read: "$\mathcal{F}^p(\hat{I}_l, \hat{I}_r) \neq \mathcal{F}^p(I_l, I_r)$, where $\hat{I}_l$ and $\hat{I}_r$ denote the stereo images with the adversarial patch $P$ applied, and $p$ indicates the corresponding pixel region of the patch within the prediction results."
> ## 5. Typo Issues.
> **W5b: Another example of poor writing is a lack of consistency. Line 33 writes "..Fig. 1..", whereas Line 39 writes "Figure 1". It would be great if these inconsistencies could be fixed.**
>
> **W9: There are multiple typos in the current manuscript, for example, ... missing space between (1242,* *375) while noting the image size for KITTI.**
>
> Thank you for pointing out these typos. We will carefully check and revise all of them in the updated version.
>
> References:
>
> [1] Attention concatenation volume for accurate and efficient stereo matching. (Xu, Gangwei, et al. CVPR 2022)
>
> [2] Improving Feature Stability during Upsampling – Spectral Artifacts and the Importance of Spatial Context. (Shashank, Agnihotri, et al. ECCV 2024)
>
> [3] Beware of road markings: A new adversarial patch attack to monocular depth estimation. (Liu, Hangcheng, et al. NIPS 2024)
>
> [4] Iterative Geometry Encoding Volume for Stereo Matching. (Xu et al., CVPR 2023)
>
> [5] Practical Stereo Matching via Cascaded Recurrent Network With Adaptive Correlation. (Li et al., CVPR 2022)
>
> [6] Unifying Flow, Stereo, and Depth Estimation.(Xu et al., TPAMI 2023)
>
> [7] Physical 3D adversarial attacks against monocular depth estimation in autonomous driving. (Zheng, Junhao, et al. CVPR 2024)
>
> [8] Vggt: Visual geometry grounded transformer. (Wang et al., CVPR 2025)

---

> ### Author Response · Authors · 2025-08-05
>
> We sincerely appreciate your thoughtful and active feedback on our rebuttal.
>
> We are pleased to know that our response has effectively addressed the main question raised in your initial review, i.e., the patch deployment setup, which is critical for ensuring a clear understanding of our core contributions. Regarding the remaining concerns, we fully agree that consistency and rigor in writing are critical for a strong submission, and we apologize for not meeting the highest standards in the original manuscript. To address the writing issues you concerned, we have carefully revised the manuscript in the following ways:
>
> 1. **Clarification on the use of “SotA”**
>
>    We have updated our descriptions of RAFT-Stereo and STTR by replacing the term “SotA” with more appropriate alternatives such as “advanced” and “pioneering.” In addition, we plan to include new experimental results in the manuscript to demonstrate the effectiveness of our proposed attack method against true state-of-the-art models.
>
> 2. **Revisions to the Related Work and Conclusion Sections**
>
>    We now understand that your concern lies in the writing not clearly aligning with the main topic and contributions of the paper. To improve clarity and focus, we have revised the relevant contents as follows:
>
>    - *Lines 74 to 80:*
>    In contrast, despite their geometric soundness and widespread deployment, stereo depth estimation systems have received relatively limited attention in adversarial research. Existing studies have primarily focused on digital, white-box attacks [1, 29], overlooking potential vulnerabilities in physical environments. This gap is particularly concerning, as stereo systems rely on precise correspondence between left and right images. Failures in such systems can lead to serious consequences, especially in autonomous applications where accurate and reliable 3D perception [32] is critical.
>
>    - *Conclusion:*
>    In this work, we present DepthVanish, a significant advancement in adversarial patch attacks that jointly optimizes both texture and interval structure to fool stereo depth estimation systems. By thoroughly analyzing the effect of interval spacing in repeated texture patterns, we introduce a novel insight into enhancing the attack effectiveness of traditional repeated textures. To demonstrate the potentially dangerous consequences of depth estimation failure, we design the patch to "disappear", where the patch is estimated as far away despite being physically close. Unlike previous methods limited to digital environments, our approach succeeds in both digital and physical settings, when tested against state-of-the-art models and commercial RGB-D cameras. These findings reveal critical vulnerabilities in current depth estimation technologies and raise concerns about their reliability in safety-critical autonomous systems.
>
> 3. **Clarification on the phrase “disappears object”**
>
>    We appreciate your concern about the phrase “disappears object,” which we agree is ambiguous. To address this, we have carefully reviewed the entire manuscript, including the supplementary material, and identified all confused parts as listed below. As the revision example shows, we now clearly state "disappears the patch" instead of expressions that potentially cuase confusion.
>
>    * Line 150
>    * Lines 270 - 271
>    * Lines 303 - 304
>    * Lines 327 - 328
>
>    *Example revision for lines 327 to 328:*
>    Unlike previous approaches that were limited to digital environments, our method effectively makes the patch disappear in both digital and physical settings when tested against state-of-the-art models and commercial RGB-D cameras.
>
> We hope that these revisions adequately address your remaining concerns. Please let us know if there are any further questions regarding our submission. We are more than willing to incorporate additional improvements.
>
> Finally, we would like to reiterate that DepthVanish is the first method to demonstrate effective adversarial attacks against stereo depth estimation systems in both digital and physical environments. This includes successful attacks on commercial stereo depth estimation systems such as Intel RealSense. Our approach is easy to implement and highly generalizable, making it a practical tool for evaluating and stress-testing the physical robustness of modern stereo depth estimation systems.
>
> Best Reagrds

---

> > ### Comment · Reviewer_ZGfn · 2025-08-05
> >
> > Thanks a lot for the response.
> >
> > I am glad to note that most of my concerns are now addressed.
> >
> > The last remaining concern is repeated overclaims, or rather, overemphasis on noteworthy things while being vague, making the statement into an overclaim.
> >
> > For instance, in the last reply, "DepthVanish is the first method to demonstrate effective adversarial attacks in both digital and physical environments". This statement is clearly not true; there are multiple prior works that demonstrate effective adversarial attacks in both digital and physical environments. For example, [1] does it for object detection, while [2] does so for "monocular" (not stereo, like in this submission) depth estimation.
> >
> > I would highly appreciate it if these overclaims were avoided.
> >
> > Best Regards
> >
> > References:
> >
> > [1] Wu, Zuxuan, et al. "Making an invisibility cloak: Real world adversarial attacks on object detectors." European Conference on Computer Vision. Cham: Springer International Publishing, 2020.
> >
> > [2] Zheng, Junhao, et al. "Physical 3D adversarial attacks against monocular depth estimation in autonomous driving." Proceedings of the IEEE/CVF Conference on Computer Vision and Pattern Recognition. 2024.

---

> > > ### Author Response · Authors · 2025-08-05
> > >
> > > Thank you for pointing out the overclaim. As you correctly noted, there are indeed several prior works that demonstrate effective adversarial attacks in both digital and physical environments, particularly in object detection and monocular depth estimation tasks.
> > >
> > > Our intention was to emphasize that DepthVanish specifically targets stereo depth estimation systems, a domain that has received comparatively less attention in the context of physical-world adversarial attacks. We have revised the relevant statements in the previous response to clearly reflect this specific scope and to avoid any unintended overstatement. To further strengthen the submission, we will carefully review and revise the main content to ensure that no such overclaims remain in the final version.
> > >
> > > We greatly appreciate your thoughtful feedback and the opportunity to address this. We hope the additional clarification resolves the concern, and please do not hesitate to let us know if there are any further issues regarding the manuscript.
> > >
> > > Best Regards

---

> > > > ### Comment · Reviewer_ZGfn · 2025-08-05
> > > >
> > > > Great, thank you for the discussion.
> > > >
> > > > My concerns have been addressed.
> > > >
> > > > I will have a look at the other reviews and responses and accordingly decide my final recommendation.
> > > >
> > > > I will come back for further discussion in case I have any further questions regarding the responses to the other reviews.
> > > >
> > > > Best Regards

---

> > > > > ### Author Response · Authors · 2025-08-06
> > > > >
> > > > > Thank you very much for your time and thoughtful feedback throughout the discussion. We appreciate your engagement with our work and are glad to hear that your concerns have been addressed. Please feel free to reach out if any further questions arise.
> > > > >
> > > > > Best Regards

---

### Note · Authors · 2025-08-13

Dear Area Chair and Reviewers,

Greetings from the authors of submission #8509. We would like to sincerely thank you for the time, effort, and valuable feedback you have provided throughout the review and rebuttal process.

In this work, we present DepthVanish which is the first method to simultaneously achieve both effective physical and digital attacks against stereo depth estimators, including successful attacks on commercial stereo depth estimation systems such as Intel RealSense. Our approach is easy to implement and highly generalizable, making it a practical tool for evaluating and stress-testing the physical robustness of modern stereo depth estimation systems. All reviewers acknowledged our work's novelty, significance, and practical impact. They consistently highlighted key strengths: effectiveness and generalizability (Reviewers ZGfn and xpZm), comprehensive methodology and experiments (Reviewer FaQu), and overall novelty and effectiveness (Reviewer aSEK).

During the rebuttal and discussion stages, we received many constructive comments, for which we would again like to thank all reviewers for the thoughtful engagement. We carefully addressed all concerns raised, including improving the quality of the writing, adding further theoretical analysis, and expanding evaluations to additional attack targets. We are pleased that Reviewers xpZm, FaQu, and ZGfn expressed satisfaction with our responses, but regret that we were unable to engage in more in-depth discussions with Reviewer aSEK during the rebuttal stage where we hope our response adequately addressed the concerns.

Given the consensus and the productive outcome of the rebuttal and discussion, we respectfully request a prudent consideration of our work. Thank you again for your thoughtful reviews and kind support.

Best regards,

The Authors of Submission #8509

---

### Decision · Program_Chairs · 2025-09-17

**Decision:**

Accept (poster)

**Comment:**

This manuscript proposes an adversarially optimized targeted patch attack for stereo depth estimation. DepthVanish optimizes adversarial patches jointly for structured interval spacing and texture using a joint optimization pipeline It is evaluated on e.g. RAFT-Stereo and STTR and commercial RGB-D cameras, i.e. for digital attacks and physical attacks.
After the discussion phase, all major concerns, mostly regarding paper writing, clarity of method presentation and experimental setup, figure resolution, etc.  are addressed., and all reviewers agree that the proposed method is interesting. Since there is no revision, the reviewers and AC strongly hope that all promised clarifications will be included in the final version.